# Early Enteral Nutrition in Paediatric Acute Pancreatitis—A Review of Published Studies

**DOI:** 10.3390/nu14163441

**Published:** 2022-08-22

**Authors:** Jan Stanisław Bukowski, Łukasz Dembiński, Marcin Dziekiewicz, Aleksandra Banaszkiewicz

**Affiliations:** Department of Paediatric Gastroenterology and Nutrition, Medical University of Warsaw, 02-091 Warszawa, Poland

**Keywords:** early enteral nutrition, pancreatology, paediatric acute pancreatitis

## Abstract

Nowadays, nutrition is said to be an integral aspect of acute pancreatitis (AP) treatment. Early enteral nutrition (EEN) is safe and beneficial for patients. This was confirmed by clinical experience and can be found in guidelines on managing adults with AP. Furthermore, paediatric recommendations encourage EEN use in AP. However, paediatric guidelines are based exclusively on studies in adults. Therefore, we present a review of published studies on the time of nutritional interventions in children with AP. A search was independently conducted in April 2022 by two of the authors. Only full-text papers published in English involving children between 0–21 were considered. Only four papers met our inclusion criteria: one randomised-control trial (RCT), one prospective study with retrospective chart review, and two retrospective chart reviews. All studies supported EEN and there was no recommendation of any delay in its initiation. The results of all four papers suggested EEN with a regular, normal-fat diet. EEN is safe in children with mild or moderately severe AP and may decrease the length of hospitalisation. Unfortunately, all the conclusions are based on a small amount of heterogeneous data that are mostly retrospective. Future prospective RCTs are needed.

## 1. Introduction

Acute pancreatitis (AP) is a reversible process characterised by inflammatory infiltration with structural and functional changes. An increasing incidence of AP in children has been observed in the past two decades and is estimated to be around 3–13 in 100,000 children per year [1].

According to Tian et al. meta-analysis the main aetiology of paediatric AP (25–29% of all cases) was idiopathic pancreatitis, which had the highest incidence in Europe, North America and South America. Trauma was the main cause (32%) in Oceania while gallstones in Asia (33%). The other factors were: systemic diseases, alcohol, medication, genetics, infections, post-endoscopic retrograde cholangiopancreatography, anatomic anomalies and metabolic diseases [2]. In adults, gallstones and alcohol misuse are the main risk factors for AP [3].

There are three types of paediatric AP: mild (not associated with any organ failure, local, or systemic complications), moderately severe AP (with either the development of transient organ failure/dysfunction lasting ≤48 h or development of local or systemic complications), and severe AP (with the development of organ dysfunction that persists >48 h) [4]. In the paediatric population, most AP cases are mild or moderately severe. The prevalence of severe AP is about 17% [2].

Treatment of the majority of children with mild/moderately severe AP is conservative and includes proper fluid therapy, management of electrolyte disturbances and appropriate pain management [5]. Early mobilization of the patient and regular bowel movements are also important. In patients with AP caused by choledocholithiasis, endoscopic treatment is needed. In our paper, we focused on nutrition as an integral aspect of AP treatment.

Nowadays, nutrition is said to be an integral aspect of acute pancreatitis (AP) treatment. Despite the lack of clinical evidence, patients with AP were historically kept *nil per os* (nothing by mouth (NPO)) to ‘rest the pancreas’. Most guidelines in the past recommended NPO until resolution of pain; some suggested awaiting normalisation of pancreatic enzymes or even imaging evidence of resolution of inflammation before resuming oral feedings [6]. Instead of enteral feeding, parenteral nutrition was recommended.

However, clinical and experimental studies have shown that bowel rest is associated with intestinal mucosal atrophy and increased infectious complications due to bacterial translocation from the gut. In the 1990s, it was discovered that enteral nutrition was as effective as and even safer than parenteral nutrition in treating AP. However, it was still believed that feeding through an enteral tube placed in the jejunum, behind the Treitz ligament, was the most appropriate method [7,8].

At the beginning of the 21st century, it was proven that intragastric nutrition was as effective but much better tolerated than intrajejunal nutrition [7,8]. The results of subsequent studies showed that early oral/intragastric nutrition was possible not only in mild or moderate AP, but also in severe AP. Furthermore, multiple studies have shown that patients administered oral feeding early in the course of AP have shorter hospital stays, decreased infectious complications, and decreased morbidity and mortality [9,10,11,12].

Moraes et al., in their prospective, randomised-controlled, double-blind clinical trial, found that early nutrition with a full solid diet with normal amounts of calories and fat throughout the refeeding period in adult AP was well tolerated and was associated with a shorter length of hospitalisation (LoH) in patients without abdominal pain relapse. However, there was no significant difference in the total LoH between the liquid, soft, and solid diet groups [13]. Larino-Noia et al., in their randomised open-label trial in adults, observed that refeeding after mild AP with a full caloric diet was safe and well tolerated [14].

Around the world, these findings have led to changes in the recommendations for nutrition in adults with AP. The European Society for Clinical Nutrition and Metabolism, in the latest guidelines from 2020, recommended oral nutrition as soon as may be clinically tolerated, independent of serum lipase activity, in patients with predicted mild AP. They recommend a low-fat, soft oral diet, considering hyperlipidaemia as the third most common cause of AP [15]. Furthermore, in 2018, the American Gastroenterological Association recommended early (within 24 h) oral feeding as tolerated, rather than keeping the patient NPO in AP cases [16].

According to Pendharkar et al. oral nutrition tolerance is defined as a lack of pain relapse, nausea, vomiting and/or pain medication escalation after the introduction of oral feeding [17]. Similar assumption presented Pouthoulakis et al., who in their very interesting paper found that oral nutrition intolerance is relatively common complication of AP in adults and is associated with worse clinical outcomes in AP. What is more, young age, male gender, alcohol use, smoking, elevated blood urea nitrogen and hemoconcentration are potential predictors of oral nutrition intolerance [18]. Unfortunately, there is very little data in the paediatric population with AP. Among the most frequent symptoms only persistent vomiting was found to influence enteral feeding. Abu-El-Haija et al. evaluated the relation of feeds on the pain severity compared with patients that were kept NPO and found no difference in the reported scores in pain severity between the groups that received feeds of any kind compared with the group that was kept NPO [19]. Moreover, from paediatrician practise we know that any nutrition interventions may be obstructed by patient’s food selectivity, lack of cooperation with patients and/or parents/guardians.

To date, due to a lack of studies on the paediatric population, existing recommendations regarding the nutritional treatment of AP in children have been based on adult data, and they are not entirely consistent; we will briefly outline these recommendations.

According to the European Pancreatic Club in collaboration with the Hungarian Pancreatic Study Group evidence-based guidelines (2018), oral feeding can be started as soon as tolerated regardless of systemic inflammation or elevated amylase or lipase activity. Enteral tube feeding is recommended in case of insufficient calorie intake within 72 h or intolerance of oral feeding. Parenteral nutrition should be used as a second-line treatment [20].

A Clinical Report from the North American Society for Pediatric Gastroenterology, Hepatology, and Nutrition (NASPGHAN) Pancreas Committee (2018) suggested that children with mild AP may benefit (shorter LoH and decreased risk of organ dysfunction) from early (within 48–72 h of presentation) oral/enteral nutrition. If enteral nutrition is not possible for more than 5–7 days, parenteral nutrition should be considered. Nevertheless, the authors suggested that enteral nutrition should be introduced as soon as possible, and a combination of enteral and parenteral nutrition was more beneficial than total parenteral nutrition. It is unclear whether enteral feeding is unfavourable in cases of AP with duct fracture or disruption [5].

The NASPGHAN Pancreas Committee and European Society for Pediatric Gastroenterology, Hepatology, and Nutrition Cystic Fibrosis/Pancreas Working Group (2018) recommended that children with mild AP should be started on a general (regular) oral diet, preferably within 48 h of admission. Moreover, they recommended that jejunal tube feeding should be reserved for cases of poorly tolerated oral or nasogastric (NG) tube placement in all patients with AP. A combination of enteral and parenteral nutrition should be used in patients who do not meet caloric goals after one week of hospitalisation [21].

As previously mentioned, these recommendations are based only on studies in adults with AP. We will present a review of the studies published thus far describing the time of nutritional interventions in children with mild or moderately severe AP.

## 2. Materials and Methods

A search was conducted independently in April 2022 by two of the authors in parallel using PubMed (MEDLINE) and EMBASE (Ovid), using ‘acute pancreatitis’ AND ‘children’ AND (‘nutrition’ OR ‘feeding’) as search terms. Only full-text papers published in English involving children between 0 and 21 years of age were considered. We focused on all types of clinical trials describing the time of nutritional interventions in children with mild or moderately severe AP. All the reference lists of the articles were manually searched to identify further relevant articles.

## 3. Results

As a result of the primary search of databases, we identified 1361 articles. In total, only four papers met our inclusion criteria: one randomised-control trial (RCT), one prospective study with retrospective chart review, and two retrospective chart reviews.

The characteristics of the included studies are presented in Table 1. In total, nutritional interventions in 394 children with AP were analysed.

### 3.1. Time of Commencement of Enteral Nutrition

The time of commencement of enteral nutrition is one of the most controversial and questioned issues in the treatment of AP.

Ledder at al. and Ellery et al., in their prospective studies, demonstrated that patient-directed early oral nutrition (within a median of 19 h and 14 h, respectively) had no effect on outcomes compared to patients with initial fasting.

Abu-El-Haija et al., in their retrospective study, compared 24 patients who were allowed to eat within 24 h of being admitted to the hospital with 14 patients who were NPO during that time. In the absence of unambiguous nutritional guidelines for AP at the authors’ institution, the decision to start the diet was left to the doctors’ discretion. The authors found that early feeds were not associated with increased pain or increased LoH in patients with AP. Therefore, they concluded that early feeds were feasible in paediatric patients with AP.

Szabo et al., in their retrospective 201-chart review of AP admissions, found that children who received feeds within the first 48 h of admission had better outcomes (i.e., shorter LoH) than NPO children.

In conclusion, all four studies supported early enteral nutrition. Moreover, there is no need to delay initiation of enteral nutrition.

### 3.2. The Route of Enteral Nutrition

Enteral nutrition may be administered orally or via NG or nasojejunal (NJ) tube. All analysed papers promoted oral intake and considered tube insertion a second-line treatment. Only one patient in Ledder et al.’s study and six children in the retrospective team-directed nutrition group in Ellery et al.’s. study required enteral tube insertion. In Szabo et al.’s study, pre-existing enteral tubes were used to provide nutrition. However, the lack of data on how many patients used the tubes made it difficult to interpret the results.

None of the authors compared routes of enteral feeding. Zhao et al., in their 2021 RCT, compared NG and NJ feeding in children with AP. They reported that the NG group had a significantly shorter time of tube feeding (10 days vs. 14 days, respectively; *p* = 0.016) and a shorter LoH (13 days vs. 16 days, respectively; *p* = 0.027) than the NJ group. The authors concluded that NG tube feeding was safe and more effective than NJ tube feeding in paediatric AP [25].

### 3.3. Type of Diet

Diet composition remains a subject of discussion. The main question is the amount of fat in early nutrition in patients with AP.

Ledder et al. offered an unrestricted diet to the early enteral feeding group and a low-fat diet was introduced after symptom resolution in the delayed enteral feeding group. They reported that early commencement of a full-fat diet was as safe as initial fasting and a subsequent low-fat diet.

Hamilton-Shield and Cusisk’s comment on Ledder et al.’s work highlighted the two-kilogram difference in weight at follow-up (median 49 days) between the two groups. They indicated that the children might have found an unrestricted diet tastier than a low-fat diet and thus ate more, which may have resulted in a decreased risk of growth and development retardation [26].

Ellerly et al., in the prospective part of their study, administered a low-fat oral diet (<5 g of fat per entree and <1 g of fat per side or snack) at the time of admission. However, treating physicians were allowed to change the diet when they deemed medically necessary.

Szabo et al. considered early enteral nutrition as a clear diet with progression to a general age-appropriate diet within 6 h, if tolerated.

Abu-EL-Haija et al., in their review, compared fat intake based on grams of fat consumed per kilogram of body weight. They claimed that a higher fat intake was associated with a significantly lower daily pain severity score (*p* < 0.001).

In summary, the results of all four studies supported early nutrition with a regular, normal-fat diet.

### 3.4. Safety of Early Enteral Nutrition

In Ledder et al.’s study, there was no difference between the groups in time to pain-free status (median 2 days in both groups; *p* = 0.95) and LoH (early nutrition group median 2.6 days vs. delayed nutrition group median 2.9 days; *p* = 0.56). Moreover, no patient from the early feeding group was readmitted, whereas two children from the initial fasting group were readmitted.

Ellery et al. found that patient-directed oral nutrition (early nutrition) was safe and decreased LoH compared with the NPO group (median 48.5 h vs. median 93 h; *p* < 0.0001).

Abu-El-Haja et al. observed no difference in pain severity (*p* = 0.21) or LoH (*p* = 0.57) between their early nutrition group and NPO group.

Szabo et al. found that early enteral nutrition (within 48 h of admission) was associated with decreased risk of complications (e.g., severe AP) (*p* = 0.0025) and transfers to intensive care units (*p* = 0.004). Moreover, it was associated with shorter LoH (*p* < 0.0001). What is more, the authors observed that early enteral nutrition with high intravenous fluid rate significantly reduced the rate of severe AP (4.2%) compared with groups kept NPO with high (17%) and low (35%) intravenous fluid rate. Nonetheless early administration of high or low volumes of intravenous fluids as a single management factor had no significant effect on the measured outcomes.

All the analysed papers suggested that early enteral feeding was safe in children with mild or moderately severe AP. Furthermore, early nutrition did not increase and, in some cases, even decreased the LoH.

## 4. Discussion

### 4.1. Early, Very Early, Immediate…

The question of when to start oral nutrition in patients with mild AP found an answer in a recently published meta-analysis of RCTs comparing the initiation of early enteral nutrition (usually after a short time of NPO followed by a gradually extended diet) with enteral nutrition started immediately after admission [27]. Five RCTs with a total of 372 patients were included in this study. It turned out that starting nutrition immediately after admission to the hospital was associated with a reduction in LoH (mean difference [MD] = 2.57, 95% confidence interval [CI] = 0.41–4.72) and a reduction in the risk of food intolerance (risk ratio [RR] = 0.78, 95% CI = 0.63–0.95) compared to early nutrition. Moreover, it reduced the likelihood of re-hospitalization after discharge (RR = 0.51, 95% CI = 0.12–2.27), the risk of disease progression to severe AP (RR = 0.76, 95% CI = 0.15–3.76), and the risk of complications (RR = 1.12, 95% CI = 0.50–2.49).

There are no similar studies in the paediatric population. However, it would be unreasonable not to adapt the above-mentioned results in everyday clinical practice in children with mild AP. Especially in the case of younger children, allowing them to freely eat selected foods and products (fresh, good-quality and age-appropriate) will contribute to an earlier start and maintenance of oral nutrition. Beyond the therapeutic effect of enteral nutrition itself, it will also improve a child’s overall well-being.

### 4.2. Type of Diet

The treatment guidelines for AP in children do not specify the foods or products used in early enteral nutrition. However, following some recommendations for adults with AP, an easily digestible, low-fat, low-fiber diet is recommended in the initial phase of enteral nutrition [15,16,28].

#### 4.2.1. Fiber

Potentially, dietary fiber in AP may have a beneficial effect by alternating the gut microbiota and reducing dysbiosis; increasing the concentration of short-chain fatty acids, especially sodium butyrate; reducing inflammation by inhibiting the production of pro-inflammatory cytokines; augmenting intestinal motility, which prevents bacterial translocation [29].

So far, no studies have been conducted in children to assess the impact of dietary fiber on the course of AP. However, in a 2015 Cochrane review, that assessed the beneficial and harmful effects of different enteral nutrition formulations in adults with AP, two trials comparing enteral nutrition enriched with fibers with a total of 103 participants were included. The first study compared a fiber-enriched diet with a polymeric one [30], and the other, with no intervention [31]. In short, patients who supplemented dietary fiber had shorter LoH and fiber supplementation did not affect the occurrence of general and local complications [32].

In a recently published single-blinded RCT, Chen et al. evaluated the effectiveness of adding a soluble dietary fiber supplement (20 g polydextrose)/day to enteral nutrition in AP. In the group of patients receiving dietary fiber supplementation, the rates of feeding intolerance were significantly reduced (59.09% vs. 25.00%, *p* < 0.05). Moreover, soluble dietary fiber was associated with decreases in the incidence of abdominal distension (72.73% vs. 29.17%, *p* < 0.01), diarrhoea (40.91% vs. 8.33%, *p* < 0.05), and constipation (72.73% vs. 12.50%, *p* < 0.001). The time to first flatus and first defecation was also significantly shorter in this group (*p* < 0.001) [33].

Earlier studies showed shorter LoH (*p* = 0.03) and reduced risk of surgical interventions (*p* = 0.05) in patients with severe AP who received oral dietary fiber supplementation [30].

#### 4.2.2. Fat

One of the most feared and controversial elements of the diet in AP is its fat content.

Guidelines for treating AP in adults recommend using an easily digestible diet in the initial phase of nutrition, despite studies in which it has been proven that the standard diet in mild AP is safe and reduces LoH [13]. Moreover, in paediatric departments, it is a common clinical practice to recommend an easily digestible diet with reduced fat content in AP.

This caution in recommending the regular diet in AP probably results from the frequent gallstone-induced pancreatitis in adults, where a diet with normal/increased fat content may induce aggravation or recurrence of abdominal pain. It should be noted that a multicentre study is planned to assess the impact of dietary fat on recurrences of AP, which is scheduled to begin in August 2022 [34].

There are no studies assessing the impact of dietary fat content on the course of AP in the paediatric population. However, the recommendations of NAGHAM from 2018 state: “Children with mild acute pancreatitis should be started on a general (regular) diet and advanced as tolerated” and specify that “as soon as feasible (preferably within 48 h of admission)”.

Concerns related to the nutritional fat content seem incomprehensible because the guidelines for both children and adults with AP recommend using industrial diets in which the fat content is not reduced.

### 4.3. Future Directions

The greatest obstacle in formulating dietary recommendations for children with AP arises from the small number of studies that can be relied upon. Therefore, these guidelines are based on the results of studies in adults. However, the different ethology of AP in the paediatric population should raise some concerns in the direct application of those recommendations to children.

For many years, there was a doctrine that patients with AP should follow a strict diet, which should be gradually expanded. The recommendation to use the diet without restrictions as soon as possible still sounds revolutionary. Historical experience shows that only the so-called hard scientific evidence can stimulate widespread change in clinical practice. For now, such evidence only exists for adult patients with AP.

There is also a need for multicentre studies evaluating nutritional interventions in children with severe AP. However, they are challenging to conduct as severe AP is very rare among children.

## 5. Conclusions

The results of studies assessing the time of nutritional interventions in children indicated that enteral, preferably oral, nutrition could be introduced as soon as possible in children with mild or moderately severe AP. In the case of insufficient oral feeding, an NG or NJ tube should be inserted. In our opinion, an NG tube is preferred, as it is easier to insert. Administering a full-fat diet is a safe and well-tolerated practice, and it may reduce the level of pain and shorten the LoH.

Unfortunately, all the conclusions in our study are based on a small amount of heterogeneous data that are mostly retrospective. Future prospective randomised controlled, multicentre studies are needed to not only evaluate the timing of introducing an oral diet and safety of feeding interventions, but also to assess the diet composition and the routes of nutrition. Special attention should be given to nutritional trials in severe paediatric AP as no studies have been conducted in this group of patients to date.

## Figures and Tables

**Table 1 nutrients-14-03441-t001:** Characteristics of the studies.

Paper	Type of Study	Patient Age (Years)	Sample Size (%)	Intervention	Differences between Groups
Ledder et al. [22]	randomised-control	2–18	15 (45%)	Group 1: NPO with intra-venous fluids followed by a low-fat diet: (1) when pain resolved and serum amylase/lipase levels decreased; (2) according to the treating physician.	(1) The median time to commencement of enteral feeding was shorter in group 2 than in group 1 (19.3 vs. 34.7 h; *p* = 0.004). (2) Earlier attainment of at least 50% of early enteral feeding (50% vs. 27% of patients on the second day; *p* = 0.19) and >75% of early enteral feeding (28% vs. 7% patients on the second day; *p* = 0.13) in group 2. Follow-up (30–60 days after the discharge): (3) weight gain (1.3 kg) in group 2 vs. weight loss (0.8 kg) in group 1 (*p* = 0.28).
			18 (55%)	Group 2: unrestricted diet immediately at a level tolerated by the patients	
Ellery et al. [23]	prospective with retrospective control	2–21	92 (75%)	Group 1: TTDN—NPO until sufficient improvement in clinical symptoms and/or biochemical markers at the discretion of the treating physician	(1) Median LoH was shorter in group 2 than group 1 (48.5 h vs. 93 h; *p* < 0.001). (2) Time to the first oral feeding was shorter in group 2 than group 1 (14 h vs. 34 h; *p* < 0.001).
			30 (25%)	Group 2: PDN—a low-fat oral diet on admission	
Abu-El-Haija et al. [19]	retrospective chart review	3–19	14 (37%)	Group 1: NPO	Mean LoH was shorter in group 2 than in group 1 (81.7 h vs. 94.7 h; *p* = 0.57).
			24 (63%)	Group 2: early enteral nutrition	
Szabo et al. [24]	retrospective chart review	0–21	50 (25%)	Group 1: NPO + low or high level of intra-venous fluids	(1) LoH in group 2 was shorter than in group 1 (2.9 days vs. 4.4 days; *p* < 0.0001). (2) The rate of severe AP was less in group 2 than in group 1 (6% vs. 24%; *p* = 0.0025). (3) The ICU transfer rate was reduced to 1.3% in group 2 from 16% in group 1 (*p* = 0.004).
			151 (75%)	Group 2: early enteral nutrition + low or high level of intra-venous fluids	

AP—acute pancreatitis; ICU—intensive-care unit; LoH—length of hospitalisation; NPO—*nil per os;* PDN—patient-directed nutrition; TTDN—treatment team-directed nutrition.

## Data Availability

Not applicable.

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
