# Peer review of "Early Enteral Nutrition in Paediatric Acute Pancreatitis—A Review of Published Studies"

_nutrients, 2022, doi:10.3390/nu14163441_

Round 1

Reviewer 1 Report

My comments:

1. The first sentence in the abstract (line 7) sounds like a conclusion, not an introduction to the topic.

2. Shouldn't you mention the symptoms of the disease that limit the possibility of feeding through the gastrointestinal tract, as it was done in the article ref. 18?

3. Is it not worth defining in the introduction what the Authors understand by the terms "mild acute pancreatitis" and "oral nutrition tolerance"?

Perhaps it is worth referring to the publication by Abu-El-Haija M, Kumar S, et al. "Management of acute pancreatitis in the pediatric population: a clinical report from the North American Society for Pediatric Gastroenterology, Hepatology and Nutrition Pancreas Committee. J Pediatr Gastroenterol Nutr 2018; 66: 159-176." also to refer to the general principles of treatment.

4. In the introduction, the authors present the recommendations of various scientific societies, which coincide with the conclusions they drew on the basis of their analysis of the literature. What is the novelty of the presented conclusions? How do these conclusions differ from known opinions on the treatment of pancreatitis in children?

5. From among 1,361 articles, 4 were selected that meet the criteria specified by the authors. Why ref. 22 was not included in the analysis since it methodologically meets the inclusion criteria.

6. Line 64 - the word "that" should be omitted.

In summary:

The publication should be extended to meet the criteria of being useful for clinical practice.

The authors point to a limited number of publications on the pediatric population. They are right. However, in "narrative review", you do not have to limit yourself to the principles that are necessary for Systematic Reviews. The authors used the literature analysis technique, which complies with the principles of a systematic review.

You have to decide on something. Narrative Review allows for a broader perspective of presenting the problem. This will help relate to published principles of clinical practice and broaden the audience to clinicians who find information in the publication possibly new to them.

Author Response

JSB

Reviewer 2 Report

Dear authors,

Your narrative review focuses on the early enteral nutrition in paediatric pancreatitis. Although introduction and discussion are well balanced, I would suggest some revisions which could improve the quality of the manuscript:

-          Introduction: a brief outline of differences between pediatric and adult aetiologies in pancreatitis may help the reader to understand differences in the clinical management

-          Methods: you should explain your decision to extend the age up to 21 year or include a brief digression in the discussion section. In the adolescent population alcoholic aetiologies significantly raises and this may influence the management

-          Discussion: please mention the fluid management, although this has proven to have an impact on some outcomes (Szabo FK, Fei L, Cruz LA, Abu-El-Haija M. Early enteral nutrition and aggressive fluid resuscitation are associated with improved clinical outcomes in acute pancreatitis. J Pediatr. 2015;167(2):397–402.e1

Author Response

JSB

Reviewer 3 Report

Recession of manuscript No. 1843621: „ Early enteral nutrition in paediatric acute pancreatitis – a narrative review, written by Jan Stanisław Bukowski, Łukasz Dembiński, Marcin Dziekiewicz and AleksandraBanaszkiewicz.

            The structure of manuscript has the commonly required criteria. The topic of presented work is very actual. Nutrition is a crucial and integral aspect of acute pancreatitis treatment. Early enteral nutrition is safe and beneficial for patients; this was confirmed by clinical experience and can be found in guidelines on managing adults with acute pancreatitis. Paediatric guidelines are based exclusively on studies in adults.

            In the present study, authors present a narrative review of studies on nutritional interventions in children with acute pancreatitis.

All the conclusions of all selected studies are based on a small amount of heterogeneous data that is mostly retrospective. Future prospective randomised controlled, multicentre studies are needed to not only evaluate the timing of introducing an oral diet and safety of feeding interventions, but also to assess the diet composition and the routes of nutrition. 

The results of all four papers suggested early enteral nutrition with a regular, normal-fat diet. Early enteral nutrition is safe in children with mild or moderately severe AP and may decrease the length of hospitalisation. The results are documented in table that present the review of the obtained data. 

            The citations are well-chosen and relevant and their format respects usual standards. The conclusion summarizes the author´s results.

Author Response

Dear Reviewer,

thank you! We would like to express our gratitude for taking the time to review our manuscript.

Yours faithfully,

Jan S. Bukowski, MD

This manuscript is a resubmission of an earlier submission. The following is a list of the peer review reports and author responses from that submission.